# Genetic Diversity of Hatchery-Bred Brown Trout (*Salmo trutta*) Compared with the Wild Population: Potential Effects of Stocking on the Indigenous Gene Pool of a Norwegian Reservoir

**Arne N. Linløkken [1],\*** , **Stein I. Johnsen [2] and Wenche Johansen [3]**

1   Faculty of Applied Ecology, Agricultural Sciences and Biotechnology, Campus Evenstad, Inland Norway University of Applied Sciences, N-2418 Elverum, Norway

2   Norwegian Institute of Nature Research, Department Lillehammer, Vormstuguvegen 40, N-2624 Lillehammer, Norway; Stein.Johnsen@nina.no

3   Faculty of Applied Ecology, Agricultural Sciences and Biotechnology, Campus Hamar, Inland Norway University of Applied Sciences, N-2418 Elverum, Norway; Wenche.Johansen@inn.no

\*   Correspondence: arne.linlokken@inn.no

**Abstract:** This study was conducted in Lake Savalen in southeastern Norway, focusing on genetic diversity and the structure of hatchery-reared brown trout (*Salmo trutta*) as compared with wild fish in the lake and in two tributaries. The genetic analysis, based on eight simple sequence repeat (SSR) markers, showed that hatchery bred single cohorts and an age structured sample of stocked and recaptured fish were genetically distinctly different from each other and from the wild fish groups. The sample of recaptured fish showed the lowest estimated effective population size $N_e = 8.4$, and the highest proportion of siblings, despite its origin from five different cohorts of hatchery fish, counting in total 84 parent fish. Single hatchery cohorts, originating from 13–24 parental fish, showed $N_e = 10.5$–19.9, suggesting that the recaptured fish descended from a narrow group of parents. BayeScan analysis indicated balancing selection at several loci. Genetic indices of wild brown trout collected in the lake in 1991 and 2010 suggested temporal genetic stability, i.e., the genetic differentiation ($F_{ST}$) was non-significant, although the $N_e$, the number of alleles per locus and the number of private alleles were lower in the 2010 sample.

**Keywords:** artificial breeding; effective population size; genetic diversity; siblings; bottleneck events; genetic structure

## 1. Introduction

Stocking artificially bred brown trout (*Salmo trutta*) is a widespread measure to compensate for reduced fish abundance due to the effects of river and lake regulations [1]. To ensure the genetic adaptation to local environment and avoiding introgression from maladapted genotypes, the breeding should be based on wild parents of local strains [2–5]. Nevertheless, artificial breeding excludes important stages of a natural life history, like mate choice, and may result in offspring of genetic combinations that are potentially diverging severely from indigenous fish. Additionally, the lack of natural selection on stages from hatching and until stocking in the wild may result in genotypes that are unfitted to the actual environment [2,6–8].

The success of stocking programs are measured either as the proportion of stocked fish being recaptured [9] or as the proportion of stocked fish in the catches [10], and this is important for both ecologic and economic considerations. Potential effects of releasing hatchery-bred fish into wild environments should also be monitored by means of genetic analysis. Low recapture rates in exploited fish stocks suggest high mortality of stocked fish, and the mortality of stocked fish is usually most pronounced soon after the release [11,12].

The mortality may in part be random, i.e., non-selective, but also potentially due to genetic based selective traits [2]. Stocking immature fish, supposed to reach the mature stage after several seasons in the wild before harvesting, will expectedly lead to the selection of genotypes that are more fit to the actual environment than the average of a hatchery brood [13]. This selection is especially important if the stocked fish prove to be reproductively successful [14,15].

The present study was conducted in the hydroelectric reservoir, Lake Savalen, in central South Norway, harboring brown trout, Arctic charr (*Salvelinus alpinus*) and minnow (*Phoxinus phoxinus*). Arctic charr were numerically dominating prior to a lake regulation in 1976, a regulation that entailed the closure of the outlet stream with its spawning and nursery areas for the lake living brown trout, leaving it to spawn only in the tributaries. Due to tapping of water (from the lake bottom) during winter, large areas of the Arctic charrs' spawning sites were left dry before hatching [16]. This led to reduced abundance of both species, and the annual yield of Arctic charr was reduced from 4.6 to 3.1 kg/ha (−20%). The brown trout yield was reduced from 1.3 to 0.45 kg/ha (−65%), and the proportion of brown trout in the estimated yield was reduced from ca. 33% in 1970 to 15% in 1990 and 2010 [16–18]. Arctic charr is known to compete with brown trout for resources [19,20], and the charr reduction led to a competitive release for brown trout. Both salmonid species are exploited by anglers, and in addition, some landowners conduct gill netting [17,18]. To compensate for the reduced yield, the regulation company and power plant owners are imposed upon to stock 6000 one-year old brown trout annually as a compensation, and these are bred from wild spawners caught in two tributaries, stripped and released into each autumn.

The aim of the study was to explore genetic diversity and structure by analyzing eight simple sequence repeats (SSR, former microsatellites) in 10 samples of brown trout. Three cohorts of hatchery-bred brown trout and one age structured sample of recaptured stocked fish were sampled and compared with samples of wild brown trout from Lake Savalen and the two tributaries, from which the hatchery fish were bred. Population genetic indices were computed and compared between samples to reveal differences between the hatchery bred and the wild fish. Hatchery fish were expected to differ genetically from the wild fish, as this was proved in an earlier survey of one of these populations including one cohort of hatchery fish [8], but comparison of different cohorts of hatchery fish was not conducted. The differentiation was explained by the limited number of parents used, and the lack of sexual selective processes in the breeding. The crucial issue in the present study was the genetic characteristics of the age structured sample of stocked and recaptured fish as compared with the wild fish. To reveal potential genetic effects of the annual stocking over 15 years, genetic diversity and effective population size ($N_e$) of two brown trout samples collected in Lake Savalen with 19 years between were compared.

## 2. Methods

### 2.1. Study Site

Lake Savalen (62.233° N 10.519° E), with a surface of 15.3 km$^2$, is situated in central South Norway and drains to the Glomma River system (Figure 1). The water level was regulated 1.0 m from 1924, and the regulation was increased to 4.70 m (702.84–707.54 m a.s.l.) from 1976, leaving approximately 27% of the lake dry during late winter [16]. Prior to the regulation of 1976, the most important spawning sites of Arctic charr were in shallow water, less than 3 m depth, and was consequently left dry at lower regulation level. The breeding brown trout for stocking was based on wild spawners caught in the tributary Mogardsbekken (62.3145° N 10.4855° E), with its tributary Sagbekken (62.3191° N 10.4841° E), the largest and most important spawning streams at present. The eggs are incubated and hatched in the Evenstad hatchery, where they are nursed for one year before pelvic fins are removed, and they are stocked in Lake Savalen (not in the tributaries) in June at length 5 to 15 cm. Annually, nine to 24 spawners were caught and in total 131 specimens were used for breeding during 2005 to 2011, and four stocked fish were caught (not used

for breeding) in the stream, i.e., comprising 3.1% of the spawners. A survey fishing in 2010 showed that stocked fish comprised 21% of the total brown trout catches and 29% of the size groups that are subject to ordinary fishery [17]. The stocked fish grew faster (ANOVA, $F_{1,105} = 31.0$, $p < 0.001$) than wild fish, possibly due to faster growth during the first year, in the hatchery (Figure 2).

### 2.2. Sampling

Sampling was performed by gill netting in Lake Savalen in 1991 [18] and 2010 [17], and by electro fishing in the tributaries Mogardsbekken in 2008 and in Sagbekken in 2008, 2011 and 2012. Samples of approximately 30 specimens were collected over stretches as short as possible (200–300 m). This was done to reduce the probability of including different subpopulations in one sample, as brown trout in streams occur in more or less differentiated subpopulations [21,22]. A sample was assumed to be representative for its population, and all samples were analyzed for kinship to reveal potential family groups which may bias the further analysis [23]. The samples were aged, based on otoliths and scales [24].

The two samples of wild fish from Lake Savalen in 1991 (SavW.91) and 2010 (SavW.10), the sample of recaptured fish (Recapt.10), and the samples from the two tributaries in 2008 (Mog.08 and Sag.08) were age structured, whereas the samples from Sagbekken in 2011 (Sag.11) and 2012 (Sag.12) were both of the 2011 cohort. The samples collected in the hatchery were of single cohorts: 2009 (Hatch.09), 2010 (Hatch.10) and 2011 (Hatch.11). The hatchery fish experienced <5% mortality after hatching, i.e., low possibility for selection from hatching to stocking.

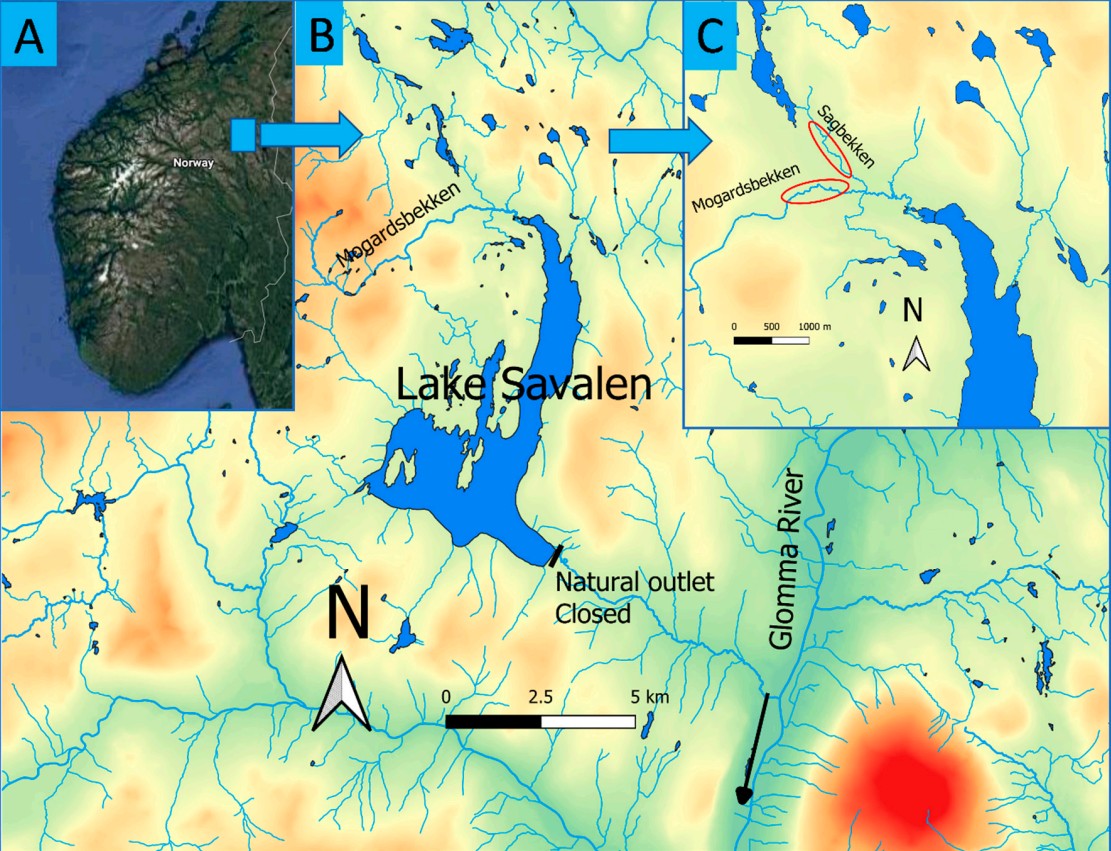

**Figure 1.** Approximately location of the study area in South Norway (**A**), overview of the Lake Savalen with tributaries (**B**), and a more detailed map of the sampled streams Mogardsbekken and Sagbekken (**C**), with red ellipses indicating the sampled stretches.

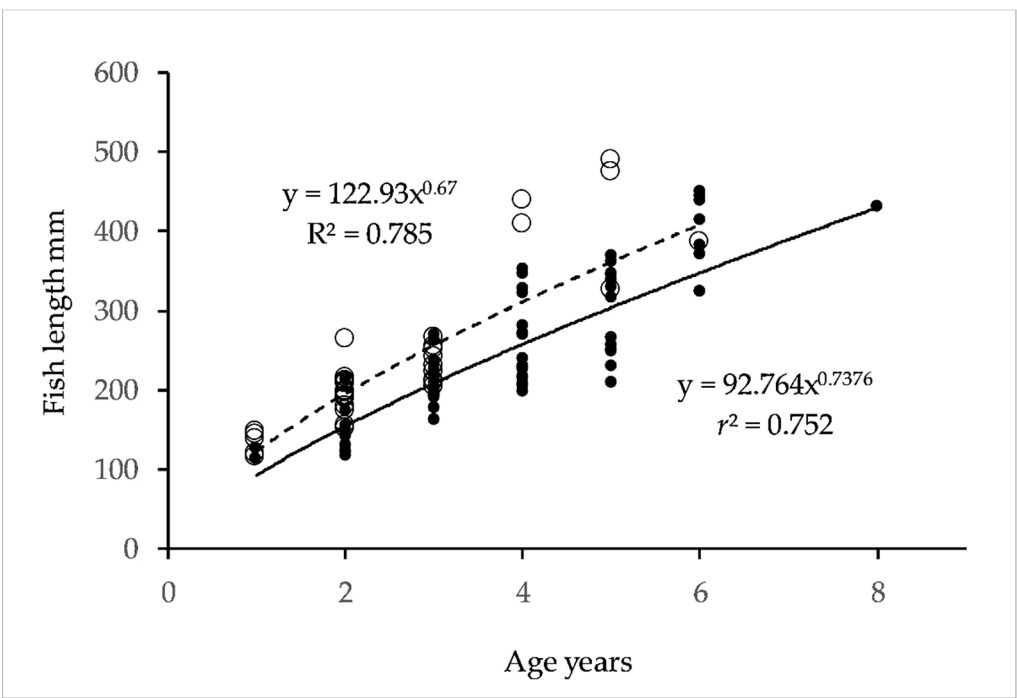

**Figure 2.** Body length of recaptured stocked (open circles, dotted line) and wild (filled circles, solid line) brown trout caught in Lake Savalen 2010, plotted on age (modified after Johnsen et al. [17]).

### 2.3. Genetic Analysis

Genomic DNA was extracted from caudal fin clips preserved in 96% EtOH, using the MagAttract DNA Blood M96 kit (Qiagen, Hilden, Germany) using a GenoM-96 Robotic Workstation (Genovision, Oslo, Norway) according to the manufacturer's instructions. An exception was the sample caught in Lake Savalen in 1991, from which DNA was isolated from scales [25]. Eight simple sequence repeats (SSRs) were PCR-amplified in two separate reactions of five (Reaction I: Ssa197 [26], SSaD170 [27], SSaD190 [27], SSaD71 [27], SSaD85 [27]) and three (Reaction II: Brun13 (=BHMS155) [28], SSa85 [26], STR73INRA [29]) markers, and both were analyzed by capillary electrophoresis (3130 XL Genetic Analyzer, Applied Biosystems (ABI)). Allele calling was performed using the GeneMapper 4.0 program (ABI). The mean scoring success across samples was 87.5%, ranging from 86.7 to 100%.

### 2.4. Data Analyses

The complete data set included ten samples with 391 individuals in total, and genotype data were converted for further analysis using the add-in utility Excel Microsatellite Toolkit (Park 2001) (Table S1), the CONVERT software [30] and the SPIDER software [31]. Number of alleles per locus ($A_L$), allele richness ($A_R$) and occurrence of private alleles ($A_P$) were calculated by means of the HP-rare software [32]. Observed ($H_O$) and expected ($H_E$) heterozygosity and inbreeding coefficient $F_{IS}$ were calculated in ARLEQUIN 3.5.1 [33] and tested for significance using a nonparametric permutation with 10,100 permutations. ARLEQUIN 3.5.1 software was also used to reveal linkage disequilibrium (LD), tested for significance with Markov chain length 1,000,000 and 100,000 dememorization steps.

Tests for deviation from Hardy–Weinberg equilibrium (HWE) were performed with the Markov chain method (with parameters dememorization 1000, 100 batches with 1000 iterations) in the web based GENEPOP [34]. Effective population size ($N_e$) was estimated using the software package NeEstimator V2 [35]. The NeEstimator software package implements the linkage disequilibrium method used by Bartley et al. [36] and calculates $N_e$ based on alleles with frequency $\geq$ 0.01, 0.02 and 0.05. Frequencies close to 0 or 1 tend to bias the estimates, and 0.02 allele frequency cut-off was chosen as recommended by Waples and

Do [37]. Parametric 95% confidence intervals were estimated. The method can be applied to age-structured isolated populations, and a minimum of eight unlinked and neutral loci are recommended. When applied to single cohort samples, the output is an estimate of the effective number of parents, or number of breeders $N_b$. The ML-Relate software [38] was run to reveal siblings within samples, and the number of probable pairs of full-siblings and half-siblings were given in percent of possible number of pairs ($n \times (n - 1)/2$).

To explore indications of recent bottleneck events, the BOTTLENECK 1.2.02 software [39] was run using an infinite allele mutation model (I.A.M.), a stepwise mutation model (S.M.M.), and a two-phase mutation model (T.P.M.), of which the T.P.M. is assumed to be the most realistic model for SSR [39]. A significant number of loci with heterozygote excess tested by means of a Wilcoxon sign-rank test indicate that a population have undergone a recent population bottleneck event. In addition, the Garza–Williamson modified index was calculated across loci by means of the ARLEQUIN 3.5.1 software. This index is a ratio $M$ = the number of alleles (k) divided by the range in allele size (r), based on the assumption that the number of alleles declines faster than the range in allele size during a bottleneck. Any data set with seven loci or more with $M < 0.68$ can be assumed to have gone through a recent reduction in size [40].

BayeScan software [41] was used to reveal possible loci under selection. The locus-population differentiation ($F_{ST}$) is decomposed into a population-specific component (beta), shared by all loci, and a locus-specific component (alpha) shared by all the populations. Selection at a locus is assumed when the locus-specific component is necessary to explain the observed pattern of diversity (alpha significantly different from 0, alpha < 0 => balancing and alpha > 0 => diversifying selection [39]). Model choice decision is performed using the so-called "Bayes factors" to choose between two models M1 (neutral) and M2 (selection), the Bayes factor BF for model M2 is given by BF = P(N|M2)/P(N|M1) [39]. The BF provides a scale of evidence in favor of one model versus another. The higher the BF, the higher the probability of selection, and "Jefferys' scale" of evidence for BF states that BF = 10–32 is interpreted as a strong probability, BF = 32–100 as very strong, and BF > 100 as a decisive probability of selection [41,42]. Tests were run with all samples included, and so were several different groups of samples, aiming to reveal potential selection comparing wild versus hatchery fish.

ARLEQUIN 3.5.1 software was used to calculate global and pairwise differentiation $F_{ST}$. Test results of $F_{ST}$ were Bonferroni-corrected for multiple testing. AMOVA was performed with the samples grouped in several ways, and $F$ statistic calculations of Arlequin express the genetic variation between groups of populations ($F_{CT}$) and between populations within a group ($F_{SC}$). The aim is to find the combination of groups giving the highest proportion of variation between groups and the lowest proportion of variation within groups.

Further, the software STRUCTURE 2.3.4 [43] was used to infer the most likely number of population clusters (*K*) constituting each sample. Each individual *i* was assigned a membership coefficient ($Q_i$) for each inferred cluster. Ten different runs were performed for each *K* (1–12, i.e., 1 − n + 2) simulated, assuming an admixture model. The following settings were used in each (120) run: The length of a burn-in period was set to 50,000, and 50,000 Monte Carlo Markov Chain (MCMC) reps were run after burn-in. The optimum number of clusters *K* was determined as described by Evanno et al. [44], and was attained by means of STRUCTURE HARVESTER software [45]. The estimated cluster membership coefficient matrices for the best fitted *K* was permuted so that all replicates have as close a match as possible using the CLUMPP software [46], and are presented in a bar plot.

The GenAIEx 6.5 [47] add in for Excel was used to calculate genetic distances as a pairwise population matrix of mean genetic distances ($\Phi_{PT}$, [48–50]) based on codominant genotypes and expressed in a pca plot (PCoA) [47] in Microsoft Excel.

## 3. Results and Discussion

### 3.1. Genetic Diversity

Analysis of eight SSR markers in the 10 samples of brown trout showed 5.0 to 6.3 alleles per loci ($A_L$) across samples of hatchery bred fish and from 6.6 to 8.0 alleles among the wild fish samples. Correspondingly, allele richness ($A_R$, for 30 individuals) ranged from 4.57 to 5.20 among the hatchery bred samples and from 5.01 to 6.73 among the wild fish (Table 1). The highest $A_L$ and $A_R$ were recorded in the two samples of wild fish from Lake Savalen (1991 and 2010), of which the 1991 sample also had the highest number of private alleles ($A_P$) of all samples, contrasting the hatchery cohorts, among which no private alleles were recorded. Arlequin analysis of linkage disequilibrium (LD) revealed LD in all pairs of loci (28) across populations, except for four pairs (14.3%) (Table S2). The LD was primarily due to the hatchery groups, of which 21–68% of the locus pairs were in LD, as compared with 3.6 to 32% among the wild fish groups. In the wild fish samples from the lake, only 3.6 and 7.1% were in LD, respectively, in the samples from 1991 and 2010. The sample of recaptured stocked fish (from here referred to as the sample of recaptures), had the highest proportion (68%) of the locus pairs in LD of all samples. After Bonferroni correction, 7.5% of the pairs were significantly in LD, of which 5.7% were among the hatchery fish and 1.8% among the wild fish samples.

**Table 1.** Genetic diversity expressed as mean number of alleles pr. locus ($A_L$), allele richness ($A_r$), private alleles ($A_P$), observed ($H_O$) and expected heterozygosity ($H_E$), estimated effective population size ($N_e$), inbreeding index ($F_{IS}$), and number of full-siblings/half-siblings in percent of possible pairs (n × (n − 1)/2) (N = sample size). * = significant deviation from HW—equilibrium, boldface = significant after Bonferroni correction.

| Sample | N | $A_L$ | $A_r$ | $A_P$ | $H_O$ | $H_E$ | $N_e$ | $F_{IS}$ | Siblings |
|---|---|---|---|---|---|---|---|---|---|
| Recap.10 | 35 | 5.3 | 4.71 | 0 | **0.75 *** | 0.70 | 8.4 (5.3–12.1) | −0.065 | 11.6/8.7% |
| Hatch.09 | 30 | 5.0 | 4.57 | 0 | 0.71 | 0.68 | 10.5 (6.8–16.4) | −0.024 | 3.9/11.7% |
| Hatch.10 | 35 | 6.1 | 5.20 | 0 | **0.68 *** | 0.67 | 19.9 (13.6–30.7) | −0.043 | 6.2/9.5% |
| Hatch.11 | 50 | 6.3 | 5.10 | 0 | **0.75 *** | 0.73 | 18.7 (14.0–25.2) | 0.016 | 5.4/9.1% |
| Pooled hatch.09–11 | 113 | 7.4 | 5.34 | 0 | 0.62 * | 0.63 | 33.0 (26.9–40.9) | −0.003 | 3.4/13.4% |
| Mean of subsamples | 35 | 6.4 | 5.40 | 0 | 0.61 | 0.63 | 28.7 (19.4–38.0) | 0.014 | 3.4/11.0% |
| SavW.91 | 33 | 8.0 | 6.73 | 5 | 0.71 | 0.77 | 481.0 (85.1–inf.) | 0.065 | 1.5/12.1% |
| SavW.10 | 34 | 7.9 | 6.53 | 2 | 0.74 * | 0.77 | 103.4 (50.4–938.6) | 0.071 | 1.8/13.9% |
| Mog.08 | 37 | 6.6 | 5.67 | 3 | 0.70 * | 0.75 | 39.5 (25.3–72.0) | 0.124 * | 2.4/14.1% |
| Sag.08 | 42 | 7.0 | 5.92 | 0 | 0.76 * | 0.75 | 38.4 (26.4–61.3) | −0.044 | 3.8/12.3% |
| Sag.11 | 50 | 6.0 | 5.08 | 0 | 0.69 | 0.67 | 43.3 (28.6–73.9) | −0.010 | 3.8/11.5% |
| Sag.12 | 47 | 6.8 | 5.46 | 0 | 0.77 * | 0.72 | 45.8 (30.5–78.2) | −0.100 | 4.3/13.0% |

Observed heterozygosity ($H_O$) ranged from 0.61 to 0.77 and was highest in the wild Sag.12, the 2011 cohort caught as a one-year old, and lowest in the mean of subsamples of pooled hatchery cohorts. There was significant deviation from HW in seven of the ten samples. $H_O$ was quite high in the sample of recaptures and in two of the single cohorts sampled in the hatchery (Hatch.09 and Hatch.11). $H_O$ was higher than expected ($H_E$), i.e., heterozygote excess, in all hatchery groups, although the test for excess was significant only for Sag.11 and Sag.12. In the pooled sample of hatchery fish, there was a heterozygote deficiency, $H_O = 0.62$ as compared with $H_E = 0.63$, and this was significantly deviant from HW (probably due to the large sample). The heterozygosity was lower in the pooled sampled as compared with the single cohort samples. In the wild fish groups, there was heterozygote deficiency except for in the three samples from Sagbekken, although the deficiency was significant only in SagW.10 and Mog.08 when tested.

The heterozygote deficiency in the lake samples probably demonstrate the Wahlund effect, caused by admixture of populations within the samples [51], and this may also explain the lower heterozygosity in the pooled hatchery sample, as compared with the single cohorts. The inbreeding index $F_{IS}$ was significant in the sample from the stream Mogardsbekken, and with the exception of the wild fish samples from the lake, the rest of the samples had negative $F_{IS}$, which is in accordance with heterozygote excess [52]. It seems clear that the artificial breeding led to reduced genetic variation, expressed as lower

$A_L$ and $A_R$, and increased levels of LD, probably due to the low number of parents used. In addition, modelling has showed that the proportion of LD in samples is higher in single cohorts as compared with age structured samples [53]. Nevertheless, heterozygosity in the first generation of bred fish was similarly to that in the wild 2011 cohort in Sagbekken. A study of brown trout of tributaries to the Lake Mjøsa, 170 km farther south, also draining to the Glomma river system, showed genetic variation (with the same markers) quite similar to those in the present study: $A_R$ = 4.38–6.19 (for 17 individuals), $H_O$ = 0.579–0.757, and $H_E$ = 0.594–0.705, and there was also a tendency to higher $H_O$ in the smallest populations as compared with the larger [54].

There were indications of recent bottleneck events in all populations according to the I.A.M., and there were indications of bottlenecks in the Mogardsbekken sample and in all hatchery groups except for the 2010 cohort, according to T.P.M (Table 2). As a rule of thumb, Garza-Williams index suggests recent population bottleneck when more than 7 loci are used in the analyses and the value is lower than 0.68. In all collections (hatchery and wild), the mean index value is lower than this reference value of 0.68 in six to seven loci. Therefore, all wild trout populations may by suffering recent bottlenecks. Bottlenecks can explain the reduction of the number of private alleles at Lake Savalen between 1991 and 2010 (Table 1). It should be expected that breeding hatchery fish from less than 30 parents causes a bottleneck effect, and hence, each hatchery cohort exhibits a founder effect, potentially with allele frequencies diverging from the source population. To maintain short-term fitness and prevent serious deleterious effects from inbreeding, Franklin [55] suggested that an effective population size of at least 50 is necessary, and a minimum effective population size of about 500 is needed to maintain genetic variability for adaptation to changing environmental conditions. Later, it has been recommended to increase the 50 and 500 limits to 100 and 1000 [56], but this is debated [57].

**Table 2.** Output of Bottleneck analysis, showing probability (*p*) of heterozygote excess of the studied samples compared with three different models (I.A.M, T.P.M. and S.M.M.), and unmodified and modified Garza–Williamson index (number of loci with modified index < 0.68 in parenthesis).

| Sample | Wilcoxon Test H Excess *p*-Values | | | Garza–Williamson Index | |
|---|---|---|---|---|---|
| | **I.A.M.** | **T.P.M.** | **S.M.M.** | **Unmodified** | **Modified** |
| 1. Recap.10 | 0.002 | 0.002 | 0.230 | 0.435 | 0.345 (7) |
| 2. Hatch.09 | 0.002 | 0.004 | 0.273 | 0.432 | 0.322 (7) |
| 3. Hatch.10 | 0.020 | 0.473 | 0.981 | 0.496 | 0.383 (7) |
| 4. Hatch.11 | 0.004 | 0.012 | 0.578 | 0.488 | 0.375 (7) |
| 5. SavW.91 | 0.002 | 0.191 | 0.770 | 0.523 | 0.389 (6) |
| 6. SavW.10 | 0.004 | 0.156 | 0.081 | 0.477 | 0.471 (6) |
| 7. Mog.08 | 0.001 | 0.021 | 0.434 | 0.519 | 0.400 (7) |
| 8. Sag.08 | 0.002 | 0.020 | 0.680 | 0.463 | 0.437 (6) |
| 9. Sag.11 | 0.018 | 0.232 | 0.010 | 0.479 | 0.393 (6) |
| 10. Sag.12 | 0.006 | 0.126 | 0.963 | 0.455 | 0.425 |

### 3.2. Effective Population Size $N_e$, Kinship and Selective Mortality

Effective population size ($N_e$) was generally low in the hatchery cohorts, ranging from 10.5 to 19.9 (Table 1), comprising 81 to 83% of number of parents used (13 to 24). $N_e$ was lowest in the sample of recaptured fish, only 8.4. This could in part be due to the fact that mixed cohort samples tend to bias $N_e$ downward [53], and this sample of 35 specimens consisted of fish from five age groups (cohorts 2005 to 2009, of which the 2007 and 2008 cohorts dominated by number, with 11 and 14 individuals, respectively, (Supplementary Table S3)). Unfortunately, samples for genetic analysis were not collected from the hatchery cohorts 2005 to 2008, but the number of parents used to breed these cohorts was in total 86, of which 34 were females. The pooled sample of the analyzed hatchery cohorts 2009 to 2011 was bred from in total 58 parents, of which 29 were females, and $N_e$ of the pooled sample was 33.0. Five subsamples of 35 specimens (to compare with the 35 recaptured

specimens), randomly drawn from the pooled sample, gave a mean $N_e$ = 28.7, i.e., a 13% reduction compared to the estimate based on the total of 58 specimens, but still more than threefold that of the sample of recaptures.

This suggested that the recaptured fish descended from a limited number of the 86 breeders used for the 2005–2009 cohorts. This conclusion was supported by the high proportion of probable full and half sibling pairs, 11.7% and 8.7%, respectively, of possible pairs, in the sample of recaptures. The number of sibling pairs otherwise varied from 3.9 to 6.2% of full sibling pairs and 9.1–11.7% half siblings, among the cohorts collected in the hatchery. In the wild fish groups, the number of probable full sibling pairs was 1.5 to 4.3%, and 11.5 to 14.1% half sibling pairs. The proportion of siblings was lowest in the samples from Lake Savalen, not unexpected as the lake stock is recruited from several spawning populations.

The stocked fish may have experienced selective mortality between stocking and recapture, favoring rather few genetic combinations or family groups. Analysis for loci potentially under selection, by means of the BayeScan software, indicated selection on all loci except at two (PO > 10), when including all samples in the analysis, whereas no selection was indicated among the samples collected in the hatchery (Table 3). When combining all groups of wild fish, the analysis revealed decisive probability (PO = 1250) of balancing ($\alpha$ < 0) selection on locus SSaD170, and when including the sample of recaptures, there was still a decisive probability of balancing selection on locus SSaD170 and a strong probability of balancing selection on loci SSaD71, SSa85 and STR73I. The balancing selection hopefully ensures that stocked fish, growing to maturation and potential participation in reproduction, was adapted to the local environment, having survived for one to four years in the lake environment. The indicated selection on several loci may affect the accuracy of the estimated $N_e$, although, the effect is assumed to be minor [37]. A formerly cited study including single nucleotide polymorphism (SNP) genotyping on wild brown trout in Sagbekken (Sag.11 and Sag.12), compared with hatchery fish, suggested selection on several SNPs, and some SNPs seemed to be related to body size, and fast growing specimens of fry seemed to be favored [8].

Assuming that stocked fish comprise a minor proportion of the spawners, harmful effects on the wild fish genetics is not likely. A proportion of >10% stocked fish with reproductive success among the spawners is shown to gradually reduce the allele richness and genetic diversity of the wild stock [58], and also reduce the effective population size [58–60]. There was a decrease in $N_e$ of the lake samples from 1991 to 2010, actually from 481 to 103, or close to 80%, although it was not significant due to large 95% CI. The suggested selection at locus SSaD170 violates the premise of the method, and the resulting $N_e$ could be biased. The genetic diversity of the brown trout in Lake Savalen, nevertheless, may be expected to decrease with continuing stocking and should be followed up by genetic analysis. Ferguson [14] reported from a review study, covering large parts of Europe, that the degree of introgression from farmed fish is highly variable, but so are the breeding methods. Some hatcheries breed from farm stocks, i.e., the breeding stock is hatchery bred. Cross breeding may lead to reduced fitness (like reduced survival and growth) [61], and according to the Norwegian Biodiversity Act, any stocking should be avoided in populations that are self-sustaining.

**Table 3.** Output table from the BayeScan analysis performed to reveal possible loci under selection. Posterior Odds (PO) is used as Bayes factors (BF) in Jefferey's scale. Bold face indicates significant probability of selection.

| Group Tested | Locus | PO | Prob. | log10(PO) | q Value | Alpha | $F_{ST}$ |
|---|---|---|---|---|---|---|---|
| Recap.10+ | SSa197 | **20.0** | 0.952 | 1.301 | 0.885 | −0.961 | 0.055 |
| Hatch.09+ | SSaD170 | **>>** | 1.000 | 1000 | 0.920 | −1.327 | 0.039 |
| Hatch.10+ | SSaD190 | 7.0 | 0.875 | 0.843 | 0.863 | −0.804 | 0.063 |
| Hatch.11+ | SSaD71 | **624** | 0.998 | 2.795 | 0.925 | −1.152 | 0.046 |
| SavW.91+ | SSaD85 | **2499** | 1.000 | 3.398 | 0.907 | −1.148 | 0.046 |
| SavW.10+ | Brun13 | 0.64 | 0.391 | −0.192 | 0.914 | −0.274 | 0.099 |
| Mog.08+ | SSa85 | **33.7** | 0.971 | 1.528 | 0.898 | −1.150 | 0.048 |
| Sag.08+ | STR73I | **>>** | 1.000 | 1000 | 0.799 | −1.740 | 0.028 |
| Sag.11+ | | | | | | | |
| Sag.12 | | | | | | | |
| Hatch.09+ | SSa197 | 0.08 | 0.070 | −1.125 | 0.916 | −0.033 | 0.078 |
| Hatch.10+ | SSaD170 | 0.05 | 0.051 | −1.268 | 0.932 | −0.011 | 0.079 |
| Hatch.11 | SSaD190 | 0.11 | 0.099 | −0.958 | 0.901 | −0.066 | 0.076 |
| | SSaD71 | 0.06 | 0.053 | −1.252 | 0.926 | 0.001 | 0.080 |
| | SSaD85 | 0.06 | 0.053 | −1.249 | 0.922 | −0.008 | 0.079 |
| | Brun13 | 0.05 | 0.052 | −1.263 | 0.929 | 0.013 | 0.081 |
| | SSa85 | 0.08 | 0.070 | −1.123 | 0.911 | −0.025 | 0.078 |
| | STR73I | 0.11 | 0.099 | −0.961 | 0.901 | −0.055 | 0.077 |
| Recap.10+ | SSa197 | 0.08 | 0.071 | −1.115 | 0.885 | −0.032 | 0.076 |
| Hatch.09+ | SSaD170 | 0.05 | 0.046 | −1.319 | 0.920 | −0.012 | 0.077 |
| Hatch.10+ | SSaD190 | 0.08 | 0.072 | −1.111 | 0.863 | −0.036 | 0.076 |
| Hatch.11 | SSaD71 | 0.04 | 0.043 | −1.352 | 0.925 | 0.002 | 0.078 |
| | SSaD85 | 0.06 | 0.054 | −1.240 | 0.907 | −0.013 | 0.077 |
| | Brun13 | 0.06 | 0.052 | −1.259 | 0.914 | 0.014 | 0.079 |
| | SSa85 | 0.07 | 0.064 | −1.164 | 0.898 | −0.027 | 0.077 |
| | STR73I | 0.25 | 0.201 | −0.599 | 0.799 | −0.219 | 0.069 |
| SavW.91+ | SSa197 | 0.43 | 0.298 | −0.372 | 0.316 | −0.260 | 0.048 |
| SavW.10+ | SSaD170 | **1250** | 0.999 | 3.097 | 0.001 | −1.327 | 0.018 |
| Mog.08+ | SSaD190 | 1.26 | 0.557 | 0.100 | 0.197 | −0.529 | 0.038 |
| Sag.08+ | SSaD71 | 3.34 | 0.769 | 0.523 | 0.136 | −0.836 | 0.029 |
| Sag.11+ | SSaD85 | 6.16 | 0.860 | 0.790 | 0.070 | −0.877 | 0.028 |
| Sag.12 | Brun13 | 0.04 | 0.040 | −1.378 | 0.397 | −0.002 | 0.060 |
| | SSa85 | 4.78 | 0.827 | 0.679 | 0.104 | −1.049 | 0.025 |
| | STR73I | 0.91 | 0.476 | −0.042 | 0.252 | −0.617 | 0.039 |
| Recap.10+ | SSa197 | 2.48 | 0.713 | 0.394 | 0.103 | −0.686 | 0.048 |
| SavW.91+ | SSaD170 | **2500** | 1.000 | 3.398 | 0.000 | −1.242 | 0.028 |
| SavW.10+ | SSaD190 | 0.93 | 0.482 | −0.031 | 0.162 | −0.417 | 0.060 |
| Mog.08+ | SSaD71 | **31.0** | 0.969 | 1.492 | 0.016 | −1.096 | 0.033 |
| Sag.08+ | SSaD85 | 5.59 | 0.848 | 0.748 | 0.066 | −0.799 | 0.043 |
| Sag.11+ | Brun13 | 0.05 | 0.052 | −1.261 | 0.260 | −0.018 | 0.083 |
| Sag.12 | SSa85 | **14.2** | 0.934 | 1.152 | 0.032 | −1.147 | 0.032 |
| | STR73I | **11.3** | 0.919 | 1.054 | 0.045 | −1.323 | 0.029 |

### 3.3. Genetic Structure

AMOVA performed in ARLEQUIN gave a significant global $F_{ST} = 0.031$ ($p < 0.001$), suggesting genetic structuring, and pairwise, $F_{ST}$ was significant between all pairs, with four exceptions: Lake Savalen 1991 and 2010, Lake Savalen 1991 and Sagbekken 2008, Lake Savalen 2010 and Sagbekken 2008, and Lake Savalen 2010 and Mogardsbekken 2008 (Table 4), i.e., wild fish samples from the lake were involved in all the non-significant tests. After Bonferroni correction, the following additional three comparisons were non-significant: Sagbekken 2008 and Mogardsbekken 2008, Sagbekken 2011 and 2012, Lake Savalen 2010 and the hatchery cohort 2011, i.e., all the non-significant corrected tests were, with one exception, between wild fish samples, and the significant $F_{ST}$ indices occurred

primarily between pairs including hatchery bred fish. The wild fish groups were less structured, and repeating AMOVA including only the wild fish groups gave global $F_{ST} = 0.020$ ($p < 0.001$), whereas exclusively for hatchery fish groups, $F_{ST}$ was 0.042 ($p < 0.001$), i.e., twice as high. Stocked fish with reproductive success may therefore potentially introduce an artificial genetic structure in the brown trout stock.

**Table 4.** Pairwise $F_{ST}$ of the ten analyzed samples. Significant tests * ($p < 0.05$), and significant after Bonferroni correction (i.e., $p < 0.001$) in boldface.

| | | Sample No | | | | | | | | |
|---|---|---|---|---|---|---|---|---|---|---|
| Sample | No | 1 | 2 | 3 | 4 | 5 | 6 | 7 | 8 | 9 |
| Recap.10 | 1 | - | | | | | | | | |
| Hatch.09 | 2 | **0.051 *** | - | | | | | | | |
| Hatch.10 | 3 | **0.036 *** | **0.081 *** | - | | | | | | |
| Hatch.11 | 4 | **0.042 *** | **0.043 *** | **0.047 *** | - | | | | | |
| SavW.91 | 5 | **0.043 *** | **0.068 *** | **0.033 *** | **0.036 *** | - | | | | |
| SavW.10 | 6 | **0.022 *** | **0.045 *** | **0.021 *** | **0.012 *** | 0.008 | - | | | |
| Sag.08 | 7 | **0.034 *** | **0.061 *** | **0.025 *** | **0.031 *** | 0.005 | 0.003 | - | | |
| Mog.08 | 8 | **0.037 *** | **0.051 *** | **0.029 *** | **0.017 *** | **0.024 *** | 0.001 | **0.014 *** | - | |
| Sag.11 | 9 | **0.059 *** | **0.112 *** | **0.030 *** | **0.051 *** | **0.037 *** | **0.037 *** | **0.033 *** | **0.042 *** | - |
| Sag.12 | 10 | **0.033 *** | **0.078 *** | **0.018 *** | **0.034 *** | **0.017 *** | **0.015 *** | **0.017 *** | **0.024 *** | 0.007 * |

Further AMOVA analysis by means of Arlequin software was performed (Table S4), dividing the samples primarily in three or four groups. The second largest differentiation between groups ($F_{CT} = 0.0122$) and the second smallest differentiation within groups ($F_{SC} = 0.0256$) was found with the following grouping (Recaptured + Hatch.09 + Hatch.10 + Hatch.11), (SavW.09 + SavW.10+ Mog.08 + Sag.08) and (Sag.11 + Sag.12), i.e., one group of hatchery bred fish, one group of age structured wild fish, and one group of the cohort 2011 sampled in Sagbekken in two consecutive years. When dividing the samples in four groups, with the recaptured fish as an exclusive group, $F_{CT}$ increased to 0.0136, emphasizing the genetic deviant sample of recaptured fish. These grouping were the only two with $F_{CT}$ significantly different from null, i.e., the hatchery fish differed substantially from the wild fish, and the single cohort samples also differed from the age structured fish groups.

STRUCTURE and STRUCTURE HARVESTER analysis gave $K = 3$ Clusters as the best number of clusters, though LnP($K$) showed a maximum at $K = 7$–8 (Figure 3). At $K = 3$, Cluster 1 dominated the wild fish samples from Sagbekken and the recaptured fish, Cluster 2 was pronounced in the samples from Lake Savalen and from Sagbekken 2008 (Figure 4). The proportion of Cluster 3 was most pronounced in the hatchery cohorts of 2009 and 2011 and in the age structured sample of recaptured fish. At $K = 8$, most populations appear highly admixed, although the proportion of Cluster 1 was pronounced in the Hatch.11 sample, and was otherwise low in the other samples, with an exception for the Mog.08. Cluster 5 was pronounced in the sample of recaptures, and Cluster 7 appeared mainly among the recaptured sample and the Hatch.09, i.e., in "artificial" gene combinations. The plot demonstrates the differences between the hatchery fish groups, and suggests similarity between the wild fish sample from the lake (SavW.91 and SavW.10), and between the two samples of the 2011 cohort from Sagbekken (Sag.11 and Sag.12).

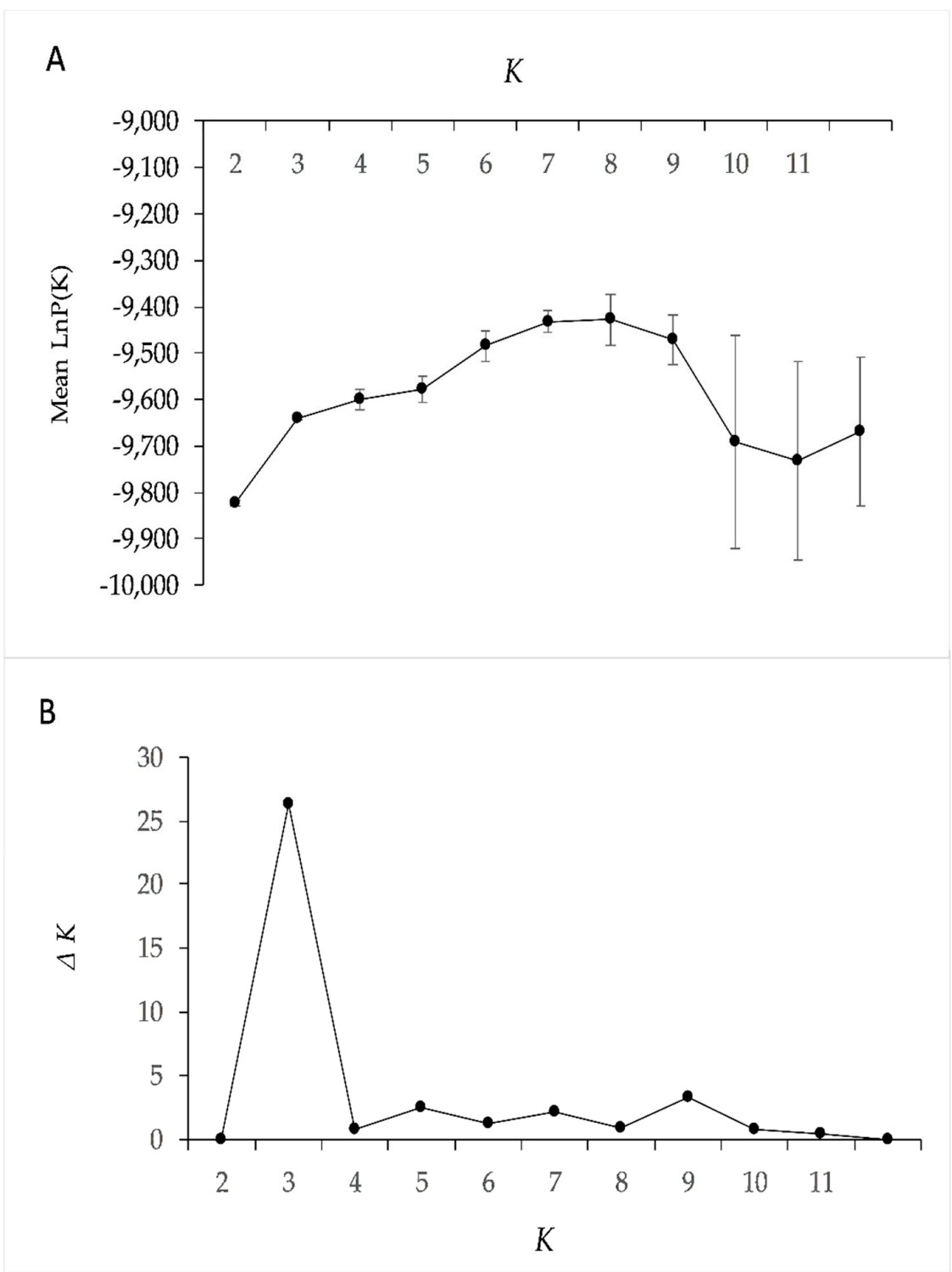

**Figure 3.** Mean LnP(*K*) (**A**) and Δ*K* (**B**) plotted on number of clusters *K* by means of the STRUCTURE-HARVESTER software.

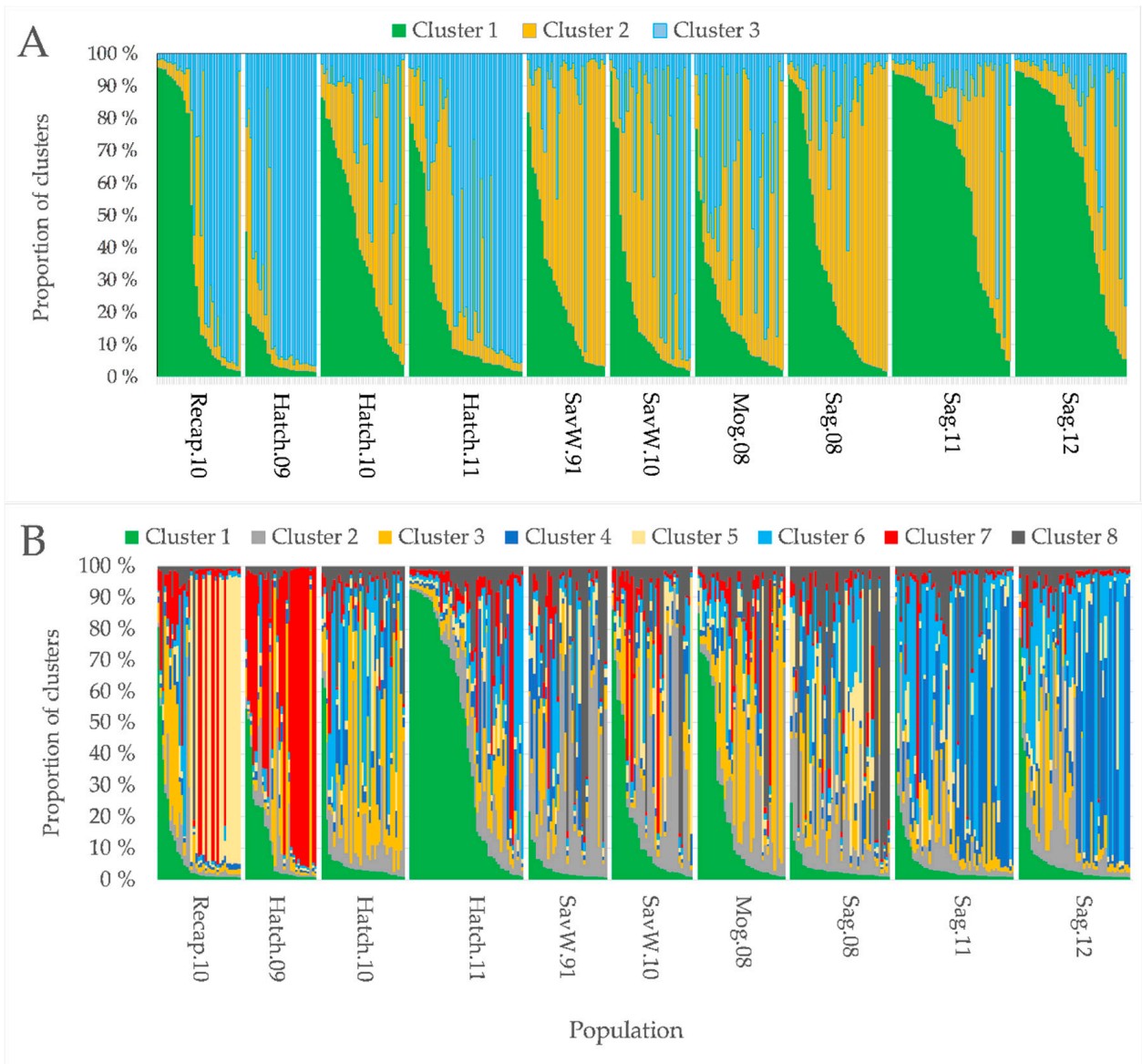

**Figure 4.** Summary plot of the estimated individual membership coefficients of each cluster 1–3 ((**A**), *K* = 3) and cluster 1–8 ((**B**), *K* = 8). Each specimen is represented by a single vertical line broken into segments, with lengths proportional to each of the *K*-inferred clusters, sorted in decaying membership of Cluster 1.

The pca plot demonstrates likewise the differentiation of the recaptured hatchery fish from the other samples, and whereas the hatchery cohort were scattered, the wild fish groups, especially the two from the lake were gathered (Figure 5).

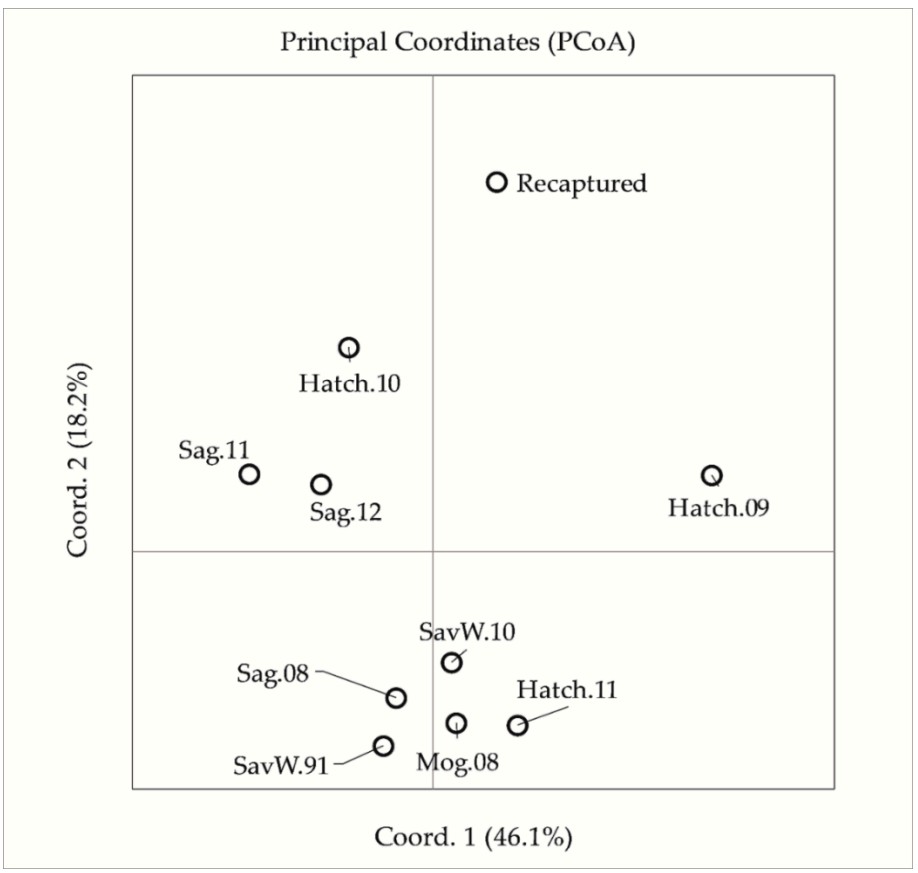

**Figure 5.** Plot of principal coordinate analysis (PCoA) based on pairwise population matrix of mean population codom genotypic genetic distance of the 10 analyzed samples. Proportion of variation explained by components Coord. 1 and Coord.2.

## 4. Conclusions

Bottleneck events due to the breeding based on low numbers of parents resulted in allele frequencies different from that of the wild fish, i.e., the source population, and there were substantial differentiation between the cohorts of hatchery bred fish, and between the hatchery fish and wild fish. A low number of randomly selected parents, forced mating, and lack of selection after hatching (low mortality) in hatchery, are plausible explanations to the genetic differentiation between the hatchery cohorts, and their differentiation from the wild fish. The age-structured sample of recaptured fish was strongly diverging from all the other groups, and despite the fact that this sample descended from five different broods and in total 86 parents, as compared with 9 to 24 parents for the single cohort hatchery samples, the sample of recaptures had the lowest $N_e$ of all. This sample also had the highest proportion of siblings, suggesting a selective mortality after stocking, favoring descendants from a few combinations of parents, possibly due to genetic fitness, as analysis indicated a strong balancing selection. Nevertheless, the breeding from new samples of wild parents each year resulted in genetically differentiated cohorts, and there was no indication of hatchery genotypes, but rather a random variation from year to year.

**Supplementary Materials:** The following are available online at https://www.mdpi.com/article/10.3390/d13090414/s1, Table S1: PopGen.3D.10.pop.allele.frequences, Table S2: Linkage disequilibrium, Table S3: Breeding stocks and recaptures, Table S4: AMOVA results from Arlequin analysis.

**Author Contributions:** Conceptualization, A.N.L.; methodology, A.N.L. and W.J.; software, A.N.L. and W.J.; validation, A.N.L., W.J. and S.I.J.; formal analysis, A.N.L.; investigation, A.N.L. and S.I.J.; resources, A.N.L. and S.I.J.; data curation, A.N.L.; writing—original draft preparation, A.N.L.; writing—review and editing, A.N.L.; visualization, A.N.L.; supervision, A.N.L.; project administra-

tion, A.N.L.; funding acquisition, A.N.L. and S.I.J. All authors have read and agreed to the published version of the manuscript.

**Funding:** This research has received economic support from the river regulation company Glommen & Lågens Brukseierforening and from Research Council of Norway.

**Institutional Review Board Statement:** The sampling of fish, with gillnetting and electrofishing, was conducted with permission and according to instructions from the Environmental Administration of Hedmark County.

**Data Availability Statement:** Our data is to be found in Supplemental Table S1.

**Acknowledgments:** We would like to thank the members of the Fishing Committee of Lake Savalen for valuable information and practical help during the sampling, and thanks also to the three reviewers for helpful and constructive criticism.

**Conflicts of Interest:** This survey was conducted in agreement with the Norwegian public environmental administration at county and national level (The Environmental Administration of Hedmark County and the Norwegian Environment Agency), as a follow-up of previously imposed surveys both before and after the lake regulation. The results of such surveys are used by the environmental authorities to evaluate the effect of imposed compensatory measures and possibly to impose new measures on developers. Both surveys and measures are paid for by the developer, i.e., in this case by the regulator Glommen & Lågens Brukseierforening and power companies. The contractors (Norwegian Institute of Nature Research and the Inland Norway University of Applied Sciences) are free to use and publish any data obtained.

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
