# Peer review of "Genetic Diversity of Hatchery-Bred Brown Trout (Salmo trutta) Compared with the Wild Population: Potential Effects of Stocking on the Indigenous Gene Pool of a Norwegian Reservoir"

_diversity, doi:10.3390/d13090414_

Round 1

Reviewer 1 Report

The manuscript by Linlokken et al analyzes the genetic impact of a permanent supportive breeding program on the brown trout, Salmo trutta, populations in the Lake Savalen (a hydroelectric reservoir).  Using just 8 microsatellite loci, this paper looks close related to a previous one by the first author (Linlokken et al, 2017) analyzing just 48 brown trout (native and hatchery reared fish) at the Sagbekken River but with a panel of up to 3700 SNPs. Authors found lower diversity levels in hatchery reared fish than in natural source populations, likely reflecting a small number of breeders collected from the rivers. In addition, significant divergence between years among hatchery reared fish contrasted with lower and even not significant divergence between brown trout populating the Mogardsbekken and Sagbekken rivers in 2008. Overall, the manuscript is interesting and well-written (but see comments below). The number of microsatellite loci looks appropriated for these kin of studies. Similarly, the number of individuals analyzed (> 300) is correct, and sample sizes were similar among collections. However, discussion should be improved and refined. In the present version of the manuscript a single, merged section is devoted to the results and the discussion, but in fact it is more a presentation of own results than an integration with available literature on stocking practices on brown trout and salmonids. Even a previous paper from Linlokken et al (2017) with similar topic and using samples here revisited for microsatellite variation is omitted. My comments and suggestions are enclosed below (in order of appearance):

  • The study area is described at section 1 in Material and Method section, but a Figure could assist to readers outside Norway to better understand the hydrographical connectivity among studied rivers.  Unfortunately the references 19 and 20 indicated in this section, are in Norwegian and therefore could hardly be understood by most of the potential readers of the manuscript. 
  • Present Table 3 should be transformed to Table 1 as information of this table is indicated in lines 120-121. In addition, the authors should indicated whether wild spawners are returned alive to the source rivers after stripping. If known, the estimated density of adult/mature fish in the source streams during the capture of these spawners could assist to validate sentence in the conclusion, lines 431-434.
  • Authors indicated (line 149 and Table 3) that a sample represents hatchery reared fish recaptured after stocking. An explanation on how this fish were identified as of hatchery origin needs to be added to the manuscript (genetic identification, clip in caudal fin, dye coloration?).
  • This sample of recaptured fish and also other three wild collections are age structured. The authors should indicated how age structured samples might affect their results, for instance on Hardy-Weinberg and Linkage disequilibrium tests.
  • Lines from 247 to 250 indicate that pairwise genetic distances were used for PCoA  but, what specific distance(s) was (were) applied? A Correspondence Analysis of individuals, as performed in the GENETIX software, may be used instead. Such analyses could identified fish moving between locations.
  • Table 1. Please indicate here what Ap (private alleles) means in the table caption.
  • Table 1. Significant and positive Fis is reported at the Mog.08 collection (in fact the largest Fis observed among studied locations/collections), indicative of heterozygote deficiency,  but this collection was in Hardy-Weinberg equilibrium. Please discuss how significant deficit of heterozygote is not detected by HWE test. Also consider revision of lines 282 to 284 because not significant Fis values were indicated in Table 1 for wild trout collections Sag11, Sag12 and SavW10.
  • Line 279. The highest Ho (0.77)  is in Sag.12 cohort not in Sag.11 (as indicated in Table 1).
  • Table 2. Revise WILCOWON TEST should be WILCOXON TEST.
  • In addition, reconsider text in lines 303 to 310. As a rule of thumb, Garza-Williams index suggests recent population bottleneck when more than 7 loci are used in the analyses and value is lower than 0.68. In all collections (hatchery and wild) the index value is lower than this reference value of 0.68. Therefore all wild trout populations may by suffering recent bottlenecks. Bottelnecks can explain the reduction on the number of private alleles at the Lake Savalen between 1991 and 2010.
  • Line 305 I.P.M should be I.A.M.
  • The sentence in lines 328-329 is somewhat confusing and refers to a circular argument because the value of the Linkage Disequilibrium is used to calculate Ne (and not Ne to inform on LD). For instance when two populations meet in a locality, there is a Wahlund effect that might produce a high value of the estimated LD among fish collected at this locality, and therefore the estimate of Ne in the locality would be small, even though both populations having a large number of breeders in their spawning streams.
  • Text in lines 355-368 related with Bayescan analyses in Table 4 is hard to follow because models (groups tested) indicated in the table are not the same as indicated in the text. Which is the difference between “Sampled in hatchery” and “All hatchery bred fish”? And between “All wild fish samples” and “Wild fish samples and recaptured”? Maybe details should be indicated in the Material and Methods section, particularly is some merging of collections are done in the comparisons. In addition, a comparison between hatchery samples and wild source locations (Mogardsbekken and Sagbekken rivers) should be added to the Bayescan analyses to check for putative changes induced at the hatchery.
  • AMOVA results in paragraph starting at line 385, was just performed to get a mean FST among locations, but because a temporal sampling at some wild locations, an additional approach could be performed in a hierarchical model where FSC estimates temporal components within locations and FSC among locations divergence. Another hierarchical model distinguishing between wild and hatchery locations could indicate the relevance of differences between hatchery cohorts and source populations (again, Mogardsbekken and Sagbekken rivers).
  • Concerning STRUCTURE results (Figure 2), an additional graphic showing the distribution of Ln(K) in front of K (also provided by STRUCTURE HARVESTER software), could be informative on the K having the highest probability, as Evanno is a conservative approach.
  • Last sentence of the conclusion is questionable. Certainly genetically differentiated cohorts are obtained each year at the hatchery, but temporal analyses among wild samples indicated low and not significant effects at lake Savalen (in a period of 19 years between samples) and between two consecutive cohorts at Sagbekken stream(2011, 2012). Weak genetic structure in the wild is indicated from FST values in Table 5 (e.g. comparisons of 2008-2010 samples in the three locations Lake Savalen, Mogardsbekken River and Sagbekken River) and distribution of STRUCTURE clusters.

Author Response

Dear reviewer

Thank you for spending time to improve our manuscript. Corrections and comments have been complied with to the best of our ability. Se attached document. Changes in the text is highlighted in yellow.

Best regards Arne Linløkken

Reviewer 2 Report

General comments:

The study has scientific value highlighting the genetic diversity and structure of hatchery-reared brown trout (Salmo trutta) as compared with wild fish in the lake and in two tributaries. Several analyzes were applied and the results are well described. However, it is necessary to improve the introduction, clarify the aims of the study, and discuss the results highlighting the main findings. I highlighted some parts of the manuscript and inserted comments bellow.

Considering the aspects mentioned above, it is necessary corrections and improvements to make acceptable to Diversity.

Specific comments:

Introduction

The introduction is very short and needs to be improved with more global approaches. Also, it should include information on abundance of the species in the lake and tributaries, a importance of the species to fishing and local economy. Are there any other studies? what are the global implications?

Lines 64-82: This paragraph must be rewrite. The hypothesis and aims of the study must be clearly highlighted. Which is the importance of the study for local context? What is expected with the study? Clarify these issues.

Line 68: hatchery

Study site

Most of these descriptions should be moved to introduction before the last paragraph.

Provide a map with localization of the sites studied.

Line 118: Are they tagged before releasing? Please, clarify it.

Lines 131-132: circles

Sampling

Line 146: check author guidelines for references – they must be numbered not cited.

Genetic analysis

Check author guidelines for references – they must be numbered not cited.

Data analysis

Check author guidelines for references – they must be numbered not cited.

Results and Discussion

Given the richness of the results obtained in the study, I was disappointed with the discussion. It is very weak, almost non-existent. So, the results must be discussed properly.

Conclusion

The conclusion looks like a discussion. So, I recommend rewriting the conclusion.

Author Response

Dear reviewer

Thank you for spending time to help us improve our manuscript. Our responses are added to the attatched document, and changes in the manuscript are highlighted in yellow.

Best regards Arne Linløkken

Reviewer 3 Report

Letter to Authors
diversity-1314717-v1
Genetic Diversity of Hatchery-Bred Compared with Wild Brown Trout (Salmo trutta) and Potential Effects of Stocking on the Indigenous Gene Pool of a Norwegian Reservoir
Arne N. Linlokken, Stein I. Johnsen, Wenche Johansen

210721

Dear authors,
This MS is potentially interesting and an important contribution to biodiversity conservation of brown trout in a Norwegian reservoir. Sampling and experimental scheme was well planned, and results were clear. It is thus worth publishing your MS in an international journal like Diversity. Before publishing your MS, you should make several points clear.
See below for detail. Words in braces are options. Bracketed words can be omitted.

L2 title
This title does not make sense.
-> {Genetic [diversity] comparison of hatchery-bred and the wild brown trout (Salmo trutta): potential effects of stocking on the indigenous gene pool of a Norwegian reservoir, Genetic diversity of hatchery-bred brown trout (Salmo trutta) compared with the wild population: potential effects of stocking on the indigenous gene pool of a Norwegian reservoir

L23
than when compared to the wild fish (redundant) -> delete

L33
two samples -> two wild samples ?

L93-100
This MS is not about the Arctic char. Condense to about a half length.

L97
after the latest regulation ??
Description of the current (latest ?) regulation in brief is necessary. I was confused to see the regulation of gill net mesh size before 1976, while no current information is presented. The essential information here might be whether mesh size is larger than before or not. Exact mesh size in L94 is not essential.

L98
What are "test fish samples"?

L186
[31] (redundant) -> delete

L217
seven loci ?
Why seven? See L167.

L240
Total number of iterations should be presented. 100,000 ?

L287
It seems clear that (weak) -> [clearly] (at an appropriate position)

L325
Effective population size Ne -> Effective population size Ne, kinship and selective mortality ?
Kinship and putative selective survival/mortality were also discussed.

L342
This conclusion was -> new paragraph

L417 Fig.3 picture
K1,K2,K3 -> cluster 1, cluster 2, cluster 3
Use of "K" is complicating with the delta-K.

L420
K1 -> cluster 1

L437
groups. -> groups, 
I think this sentence (L434-439) is too long. 

L472
Did funding from an electric power plant company induce conflict of interests?
You should write clearly what the "Glommen & Lagens Brukseierforening" is.

L474 references
Check the reference list carefully again from the beginning. Reference lists are frequently hotbeds of errors. You might add, omit or swap citation in the main text on the way internal revision. Numbering of the references might then shift. If so, readers think you are making irrelevant citation. It is the authors' responsibility that all references are properly cited.

L475,etc (many)
Make sure if paper titles are in Roman, whereas journal title words are abbreviated in Italics.

L476,etc (many)
Make sure if paper titles are in Roman lower case.

L479,etc (many)
Make sure if all authors are listed.

L480
journal? volume? pages?

and more ..

Follow the journal style written in the author guideline.

Author Response

Dear reviewer

Please find attached your comments, with authors' responses.

Best regards Arne Linløkken

Round 2

Reviewer 1 Report

This revised version of the manuscript by Linlokken et al incorporate most of my suggestions to the earlier version. I only have few minor comments that need be checked before acceptance of the paper. The first one related to the Hardy-Weinberg tests, several changes are introduced in the revised version, involving sample Mog.08 and other wild samples. Using the supplementary Table s1 with the genotype information supplied by authors for all samples and the GENEPOP on the web software, I have undertaken exact tests for HWE, heterozygote deficit and heterozygote excess (using default parameters in Markov chains), and adjusting significance levels for ten samples (p-adjusted = 0,05/10 = 0,005).  The results are presented below:

Computed p-value

Sample

HWE test

Heterozygte deficiency

Heterozygote excess

Recap.10

1.45 e-09

0.7975

0.2025

Hatch.09

0.041041

0.5936

0.4064

Hatch.10

1.42 e-07

0.0536

0.9464

Hatch.11

3.24 e-09

0.8181

0.1819

SavW.91

0.061103

0.1121

0.8879

SavW.10

0.216760

0.0063

0.9937

Mog.08

0.037907

0.0484

0.9516

Sag.08

0.044904

0.4806

0.5194

Sag.11

0.069657

0.9901

0.0099

Sag.12

0.046614

0.9777

0.0223

HWE significances are retained at Recap.10, Hatch.10 and Hatch.11 samples after Bonferroni, and no clear evidences for deficiency or excess of heterozygotes were observed. These results are clearly distinct from those presented at Table 1. The authors need to check for results in this table and revise related text in lines 253-260. If these HW disequilibrium at cohort samples from the hatchery are maintained after the checking of the results, authors should discuss on its origin.  In addition, Table 1 presented diversity indexes for pooled hatch.09-11 that are above (At and Ar) or below (Ho and He) observations at each cohort. These results are somewhat surprising and probably need some clarification within the text. The second comment related to population structure results, AMOVAS and PcoA indicate that four or 5 groups of samples are also possible to the 3 suggested by Evanno’s approach on STRUCTURE software results. In addition, the distribution of LN(K) among K, reaches a maximum at 7 or 8 cluster. Maybe authors can provide results of cluster distribution for these other K values. At any case, the text in paragraph 382-292 needs revision. Certainly hatchery and wild fish look as different buts also differences are observed among wild samples (sag.11 and 12 are different from sag. 8). Finally, check spelling throughout the manuscript, for instance:

Line 61 Hatchry should be hatchery

Line 100 whreas should be whereas

Line 218: runs performed were 10 for each K value, from K=1 to 12 (12x 10=120 no 110 as indicated within parentheses)

Line 266 “s” should be “was” (or “is”)

Author Response

Please find our response attached

Reviewer 2 Report

The authors conducted an improvement in the manuscript. So, I recommend it to publication at Diversity.

Author Response

There was no further comments from rev. 2
